# Seasonal Occurrence of Aflatoxin M1 in Raw Milk during a Five-Year Period in Croatia: Dietary Exposure and Risk Assessment

**DOI:** 10.3390/foods11131959

**Published:** 2022-07-01

**Authors:** Nina Bilandžić, Ines Varga, Ivana Varenina, Božica Solomun Kolanović, Đurđica Božić Luburić, Maja Đokić, Marija Sedak, Luka Cvetnić, Željko Cvetnić

**Affiliations:** 1Laboratory for Residue Control, Department of Veterinary Public Health, Croatian Veterinary Institute, Savska Cesta 143, 10000 Zagreb, Croatia; varga@veinst.hr (I.V.); varenina@veinst.hr (I.V.); solomon@veinst.hr (B.S.K.); bozic@veinst.hr (Đ.B.L.); dokic@veinst.hr (M.Đ.); sedak@veinst.hr (M.S.); 2Laboratory for Mastitis and Raw Milk Quality, Department for Bacteriology and Parasitology, Croatian Veterinary Institute, Savska Cesta 143, 10000 Zagreb, Croatia; lcvetnic@veinst.hr; 3Croatian Veterinary Institute, Veterinary Institute Križevci, Zakmardijeva 10, 48260 Križevci, Croatia; cvetnic@veinst.hr

**Keywords:** aflatoxin M_1_, cow milk, public health, dietary exposure, seasonal exposure, risk assessment, Croatian regions

## Abstract

This study’s objective was to estimate the seasonal occurrence of aflatoxin M_1_ (AFM_1_) in cow’s milk between winter 2016 and winter 2022 and to assess dietary exposure and risk assessment for the adult Croatian population. In total, 5817 cow milk samples were screened for AFM_1_ concentrations using the enzyme immunoassay assay (ELISA). For confirmation purposes of AFM_1_ concentration above the European Union maximum permitted level (MRL), ultra high-performance liquid chromatography with tandem mass spectrometry was performed. In 94.7% of milk samples, AFM_1_ levels were below the detection limit (LOD) of the ELISA test. For 3.47% of samples, the AFM_1_ was between the LOD and MRL values. Only 1.87% of all samples exceeded the MRL. The mean value of elevated AFM_1_ in different seasons ranged between 59.2 ng/kg (autumn 2017) and 387.8 ng/kg (autumn 2021). The highest incidences of positive AFM_1_ were determined in autumn and winter and the maximum (6.4%) was in winter 2019/2020. The largest percentage of positive samples (69.7%) was found in central Croatia. The estimated daily intakes for positive samples ranged between 0.17 and 2.82 ng/kg body weight/day. Risk assessment indicated a high level of concern during autumn and winter, especially for consumers of large amounts of milk.

## 1. Introduction

Milk is one of the most important components of the human diet, so great attention is paid to quality control and checking for possible contamination. The most important toxin in milk and dairy products is aflatoxin M1 (AFM_1_), a hydroxylated metabolite excreted in the milk of dairy animals after ingesting feed contaminated with aflatoxin B1 (AFB_1_) [1]. It is particularly dangerous because it is thermally stable, i.e., high temperatures in processing milk and dairy products cannot completely inactivate this aflatoxin [2].

Control of aflatoxins in food and feed is one of the most important issues in ensuring food safety precisely because of their negative impact on human health, since they have the highest acute and chronic toxicity of all mycotoxins [3]. The most toxic mycotoxins, aflatoxins, formed as secondary metabolites of the *Aspergillus* species are genotoxic and carcinogenic substances that can suppress the immune system and cause hepatocellular carcinoma that can cause mortality in humans and livestock [4,5]. The main risk factor for the development of hepatocellular carcinoma (HCC) has been found to be exposure to aflatoxins over a long period of time [4]. Initially, the International Agency for Research on Cancer (IARC) classified AFM_1_ into category 2B (possibly carcinogenic to humans) after it was found to have 10 times less carcinogenic potential than AFB_1_ [6]. However, given the established effects on human health, the IARC concluded there is sufficient evidence of direct carcinogenic effects of AFM_1_ and it has been reclassified as a Group 1 substance [7].

As AFM_1_ is both genotoxic and carcinogenic, it cannot be considered that there is a level of intake without any potential health hazards [8]. In order to protect consumer health, maximum permitted levels of AFM_1_ in milk have been set. The prescribed maximum permitted levels of AFM_1_ in milk vary significantly from 50 ng/kg in the European Union [9] to 250 ng/kg in Serbia [10] and 500 ng/kg in Brazil, USA, China, and Russia [11,12].

Various factors, primarily environmental, affect the increase in aflatoxins, namely drought, elevated temperatures, pest damage, biological susceptibility of the host to infection, and the potential of fungi to produce aflatoxins [13]. The occurrence of fungal infection and aflatoxin contamination can occur after ripening of crops exposed to high temperatures and moisture levels, but also after harvest and during transport, storage, processing, and handling [5].

Tropical and subtropical regions are characterised by having climatic conditions with high temperatures and long periods of drought that favour the development growth of the toxigenic mould species *Aspergillus* [14,15,16]. However, similar climatic characteristics, long periods of high temperatures, and long-lasting drought in summer have been recorded in European countries with moderate climate in the last decade, otherwise described as moderately warm humid climates with warm summers [17,18]. Such climate change has affected the increased incidence of elevated AFB_1_ in dairy cow feed [18] and appearance of elevated AFM_1_ concentrations in the milk and dairy products which was reported in countries such as Croatia [19,20], Serbia [21], Kosovo [22], and Macedonia [23].

Given the fact that AFM_1_ is a carcinogenic Group 1 compound, it is crucial to conduct constant control of its occurrence in milk and a thorough risk assessment in order to ensure the safety of milk and dairy products to protect consumer health. Therefore, the aim of this study was to estimate the occurrence of AFM_1_ in cow’s milk during different seasons in a five-year period in the territory of Croatia. Human exposure to AFM_1_ and a risk assessment with regard to milk consumption in Croatia was also conducted.

## 2. Materials and Methods

### 2.1. Sample Collection

In total, 5817 raw milk samples were collected in the period from winter 2016 to winter 2022. During that period, the laboratory received official samples of raw milk and raw milk from milk processing plants and farms sent within the self-control plans from all over Croatia. According to geographic information, milk samples were sorted by sampling area in four geographical regions in Croatia (Figure 1) as well as according to the sampling season. Milk samples were collected in sterile, 0.5 L plastic bottles and kept at 2–8 °C during transport to the laboratory, and were kept at −20 °C until analysis.

Prior to analysis, milk samples were defrosted and centrifuged for 10 min at 3500× *g* at 10 °C. The upper cream layer was removed by aspirating through a Pasteur pipette. Skimmed milk was used directly in the test (100 μL per well).

### 2.2. Chemicals and Equipment

AFM_1_ concentrations were measured using a competitive enzyme immunoassay kit (ELISA) cat. No. R1121 (R-Biopharm AG, Darmstadt, Germany). The test kit contained the following reagents: AFM_1_ standard solutions in milk buffer (0, 5, 10, 20, 40 and 80 ng/L); anti-aflatoxin M_1_ antibody (concentrate), conjugate (peroxidase conjugated AFM_1_, concentrate), substrate/chromogen (tetramethylbenzidine), stop solution (1 N H_2_SO_4_), sample dilution buffer, conjugate, antibody dilution, and washing buffer for the preparation of 10 mM phosphate buffer (PBS, pH 7.4) containing 0.05% Tween 20. Conjugate and antibody concentrates were diluted at 1:11 with the dilution buffer before analysis. Buffer salt was dissolved in 1 L distilled water and was ready for use for 4–6 weeks.

The AFM_1_ standard was purchased from Sigma-Aldrich (St. Louis, MO, USA). Aflatoxin B_1_ (AFB_1_) internal standard was used from another laboratory. The preparation of standard stock and working solutions was as previously described [19].

Immunoaffinity columns (IAC) VICAM Afla M1^TM^ HPLC were supplied by VICAM (Milford, CT, USA). Acetonitrile LC grade was purchased from Merck (Darmstadt, Germany). Ammonium formate (97%) and formic acid (≥96%) were purchased from Sigma Aldrich Chemie GmbH (Taufkirchen, Germany). Nitrogen 5.0 and 5.5 were purchased from SOL spa (Monza, Italy). Ultrapure water was obtained using the Direct-Q^®^ 5 UV Remote Water Purification System (Merck KGaA, Darmstadt, Germany).

Instrumentation used for the preparation of milk samples for the enzyme immunoassay assay (ELISA) method were: Vortex Genius 3 (IKA^®^-Werke GmbH & Co., KG, Staufen, Germany) and centrifuge Rotanta 460R (Hettich GmbH & Co., KG, Tuttlingen, Germany). Optical density was measured at 450 nm using Sunrise Absorbance Reader (Tecan Austria GmbH, Salzburg, Austria).

For ultra high-performance liquid chromatography with tandem mass spectrometry analysis (UHPLC-MS/MS), samples were prepared with the following equipment: Iskra ultrasonic bath (Ljubljana, Slovenia), IKA^®^ Vortex model MS2 Minishaker (Staufen, Germany), Supelco vacuum manifold (Bellefonte, PA, USA), centrifuge Rotanta 460R (Hettich Zentrifugen, Tuttlingen, Germany), and Nitrogen evaporation system N-EVAP^®^ model 112 (Organomation Associates Inc., Berlin, MA, USA).

An ultra high-performance liquid chromatographer with tandem mass spectrometer, UHPLC-MS/MS system consisting of UHPLC 1290 Infinity II and Triple Quad LC/MS 6470A mass spectrometer (Agilent, Palo Alto, CA, USA) was used for the analysis of samples with elevated AFM_1_ levels.

### 2.3. Analytical Determinations

AFM_1_ concentrations were measured using the ELISA screening method and for samples with concentrations above 50 ng/kg, a confirmation method using high-performance liquid chromatography with tandem mass spectrometry was used. Both analytical methods are accredited according to requirements of EN ISO/IEC 17025:2017.

Analysis of AFM_1_ was performed by the ELISA method according to the test kit instructions [17]. The method was validated according to the European Commission guidelines laid down in Commission Decision 2002/657/EC [24]. Validation parameters were reported previously and a limit of detection (LOD) and limit of quantification (LOQ) of 22.2 ng/kg and 34.2 ng/kg were determined [19]. The quality of results was tested in each assay using a negative milk sample (blank) and milk sample spiked with AFM_1_ at a known level (50 ng/kg). Quality control of the method results is checked every two years through participation in proficiency test organised by Test Veritas (Padova, Italy) and scored satisfactory results with z ≤ 2.

For confirmation purposes, the AFM_1_ concentration was determined using the UHPLC-MS/MS system. The procedure includes steps of extraction and IAC clean-up. In brief, milk samples (100 mL) were defatted by centrifugation at 3000× *g*, 4 °C for 10 min. Purification using IAC columns was started by attached it to the vacuum manifold. A 10 g sample of defatted milk was passed through the column at a rate of 2.5 mL per minute. Columns were then washed twice with 10 mL of distilled water. AFM_1_ was eluted with 2.5 mL acetonitrile at a rate of 0.5 mL per minute. The sample eluate was collected in the tube and in this step, the internal standard AFB_1_ was added. The eluate was evaporated to dryness with nitrogen at 50 ± 5 °C and dissolved with 100 μL ultrapure water and 100 μL acetonitrile (vortexed and left in an ultrasonic bath for 5 min). Samples were further centrifuged at room temperature, for 10 min at 4500× *g* and filtered through 0.22 μm PVDF filters (Agilent Technologies, Santa Clara, CA, USA) prior to injection into the UHPLC-MS/MS system.

Chromatographic separation of positive milk samples was achieved by isocratic elution using a Zorbax Eclipse Plus C18 Rapid Resolution HD, 3.0 × 50 mm, 1.8 µm particle size (Agilent Technologies, Santa Clara, CA, USA). The mobile phase for chromatography consisted of: A—5 mM ammonium formate in water with the addition of 0.1% formic acid; B—0.1% formic acid in acetonitrile. Chromatographic conditions were 60% mobile phase A and 40% mobile phase B, injection volume 7 μL, mobile phase flow 0.5 mL/min, run time 2 min, column temperature 40 °C.

The triple-quad mass spectrometer consisted of a Dual AJS ESI ion source and was operated in positive polarity with the following settings: gas temperature 350 °C, gas flow 11 L/min, nebuliser 20 psi, sheath gas temperature 300 °C, sheath gas flow 6.4 L/min, capillary voltage 5000 V, nozzle voltage 0 V and EMV 250. A multiple-reaction monitoring approach (MRM) was used for the obtained data by selecting the two most intense ion transitions of the analyte (Table 1).

The method was validated according to Commission Decision 2002/657/EC (European Commission, 2002) and the previously described procedure [20]. Recovery was calculated from the matrix-matched calibration and ranged between 100.6 and 102.1% for the four concentration levels from 0.1 to 0.75 µg/kg. These are within the limits laid down by Commission Regulation 401/2006 [25]. Satisfactory values for precision and intra-laboratory reproducibility were achieved. Relative standard deviations (RSD, %) of intra-laboratory reproducibility were lower than 18.3%. The results indicate that the UHPLC-MS/MS method used was reliable for the quantification of AFM_1_ in milk and met the criteria for detecting residues of AFM_1_. The limit of detection (LOD) and limit of quantification (LOQ) were calculated as (ng/kg) LOD 2.8, LOQ 11.0. The chromatograms of spiked cow milk and positive milk sample are shown in Figure 2.

The laboratory participates in external quality control, proficiency tests (PT), organised also by Test Veritas (Padova, Italy). The results of analyses conducted on lyophilised milk during the years showed z scores in the acceptable range −2 ≤ z ≤ 2.

### 2.4. Dietary Exposure and Risk Assessment

The exposure assessment based on determined AFM_1_ concentrations through milk consumption was conducted by calculating the estimated daily intake (EDI) using the Equation [26]:EDI = (C × P)/M,(1)
where C is the mean concentration of AFM_1_ in milk (ng/kg), P is the milk meal consumption (milk g/day), and M is the mass of the individual (kg) for adults with a body weight (BW) of 70 kg. According to the survey on food consumption conducted on the general adult population in Croatia, the mean values of chronic milk consumption for consumers are 216.2 g/day and 2.96 g/kg BW/day. For consumers consuming large quantities of milk, i.e., the worst possible exposure scenario, the value of 95th percentile (P95) was used: 511.8 g/day; 7.27 g/kg BW/day [27].

AFM_1_ concentrations below the LOD were not used to calculate exposure, i.e., they were considered inappropriate [28]. In other words, the calculation was made for a mean concentration above the EU MRL, for all milk samples (all seasons) with AFM_1_ concentrations between the LOD and EU MRL value, and for the mean concentration of the mean of all samples (all seasons) > LOD.

The estimation of the hazard index (HI) was calculated by a comparison of EDI values with the tolerable daily intake (TDI); HI = EDI/TDI [29] which is based on the proposal of Kuiper-Goodman [30]. The proposed TDI value for AFM_1_ of 0.2 ng/kg bw day^−1^ was obtained by dividing the TD50 (dose threshold by body weight) with an uncertainty factor of 5000 and is equivalent to a risk level of 1 per 100,000 [30]. The risk assessment is based on the calculation of HI where a level of HI > 1 indicates a risk for consumers.

### 2.5. Statistical Analysis

Statistical calculations were performed using Small Stata 13.1 (StataCorp LP, College Station, TX, USA). AFM_1_ concentrations (ng/kg) were expressed as the number of samples (percent) in the categories: <LOD, LOD—49.9, ≥50. Positive AFM_1_ concentrations (≥50 ng/kg) were expressed as the arithmetic mean ± SEM, minimum and maximum value. The Shapiro–Wilk test was applied to determine the distribution of the data. Statistically significant differences (*p* < 0.05) between the same seasons between different years as well as between seasons in total were analysed by the Kruskal–Wallis test.

## 3. Results

### 3.1. Occurrence of AFM_1_ in Milk

AFM_1_ was analysed in a total of 5817 milk samples collected in Croatia during the five-year period between winter 2016/2017 and winter 2021/2022. Occurrence and distribution of AFM_1_ in raw cow milk samples collected in Croatia are shown in Table 2. Overall, the determined AFM_1_ concentrations were below the LOD value in 94.7% of milk samples. For 3.47% of samples, the AFM_1_ level was between the LOD and MRL values with a mean of 33.1 ± 7.47 ng/kg. Only 1.87% of all samples exceeded the MRL value of 50 ng/kg in the range of 50.3 ng/kg and the maximum value of 1100 ng/kg measured in autumn 2021. For total milk samples with concentrations above the LOD, the AFM_1_ mean was 61.4 ± 99.5 ng/kg. The mean values of elevated AFM_1_ concentrations (>50 ng/kg) ranged from the lowest 59.2 ng/kg (autumn 2017) to the highest 387.8 ng/kg obtained in autumn 2021. Elevated AFM_1_ values were not detected during spring 2017, 2018 and 2020, and summer 2018. The order of the mean positive values with respect to the total values for the seasons was (lowest to highest) spring > summer > winter > autumn.

Statistical analysis showed significant differences (*p* = 0.0313) in AFM1 concentrations for autumn seasons between the different years. However, no statistically significant differences (*p* > 0.05) were found for other three seasons between the years. There were also no significant differences between the seasons considering their total values throughout the study period.

Analysis of positive AFM_1_ with respect to their incidence according to the seasons showed a higher number in autumn and winter periods within the ranges 0.8–4.32% and 0.91–6.4% with the highest incidence of 6.4% in winter 2019/2020. During 2019, positive samples were found in spring and summer with an incidence of 3.5% and 4.2%. Overall, considering the season, the highest incidence of 45.9% was determined in the winter periods, followed by autumn with 26.6%.

Territorial incidence of positive raw cow milk samples collected in Croatia between winter 2016/2017 and winter 2021/2022 are presented in Table 3. Positive milk samples (≥50 ng/kg) were associated with one of the four geographical regions in Croatia: Central Croatia, Eastern Croatia (Slavonia and Baranja), Croatian Littoral (Istria and Kvarner Islands) and Mountainous Croatia (Gorski Kotar and Lika) (CL-MC region), and Southern Croatian (Dalmatia) (Figure 1). The largest percentage of positive samples of 69.7% was found in central Croatia, and then in eastern and southern Croatia with similar percentages (13.8 and 14.6%). Considering the seasons in central Croatia, the highest percentage of positives, 48.7%, was found in the winter months, while 21.1% in the autumn. Given the year of observation, 44.7% of all positive samples in central Croatia were determined in 2019.

### 3.2. Exposure Assessment

Based on the mean concentration of positive AFM_1_ milk samples measured between winter 2016/2017 and winter 2021/2022, exposure and health risk were assessed (Table 4). The results revealed that the estimated daily intake (EDI) of AFM_1_ through mean milk consumption ranged between 0.10 ng/kg BW/day (for milk samples with AFM_1_ between LOD and 49.9 ng/kg) and the highest value of 1.15 ng/kg BW/day. The highest EDIs were calculated for autumn 2020 and 2021 (0.62 and 1.15 ng/kg BW/day) and winter 2021/2022 (0.62 ng/kg BW/day). EDI for total milk samples (all seasons) with concentrations above LOD was 0.18 ng/kg BW/day.

For consumers who consume large amounts of milk (95th percentile milk consumption), the highest EDI values were obtained in the range 0.24 ng/kg bw/day (AFM_1_ between LOD and 49.9 ng/kg) and 2.82 ng/kg bw/day found also for autumn 2021.

Calculated HI values below 1 were found only for the mean value of milk samples with AFM_1_ between LOD and 49.9 ng/kg and total mean (all seasons) for AFM_1_ values above the LOD, and further for winter 2016/2017 and 2017/2018, autumn 2017 and summer 2021. For all other seasons, HI was above 1, between 1.04 and 5.74. According to the highest EDIs calculated, the highest HI values, both for mean and for P95 percentile milk consumption, were determined for autumn 2020 and 2021, and winter 2021/2022: 3.10 and 7.71, 5.74 and 14.1, 3.1 and 7.62, respectively.

## 4. Discussion

### 4.1. Occurrence of AFM_1_ in Milk

In this study, only 1.87% of the total number of samples analysed in the five-year period had values exceeding the limits prescribed in the EU. The incidence of increased AFM_1_ concentrations was determined in autumn (total 26.6%) and winter (total 45.9%) periods with the highest incidence (6.4%) observed in winter 2019/2020. The obtained elevated AFM_1_ results in the autumn and winter months in the present study indicated the influence of the seasons on the occurrence of AFM_1_ contamination. This was particularly emphasised in the autumn months in which contamination of milk with AFM_1_ showed the highest mean values and statistically significant differences of means between years. This has previously been observed, i.e., the summer months show lower percentages of AFM_1_, while the winter months have a significantly higher incidence of contamination [31,32]. Seasonal variation can be explained by increased use of concentrated foods with higher amounts of mixed complementary foods such as dry hay and corn due to the reduced availability of fresh green feed in colder periods, thereby increasing exposure to feed contaminated with AFB_1_.

However, a significant impact on these results can be attributed to climatic factors or climatic extremes that affect the occurrence and growth of toxic moulds before and during maize harvest, or to the elevated temperatures and droughts present in all seasons in two regions, central and southern Croatia, especially central Croatia. Namely, the European continent, just as the rest of the world, is exposed to significant climate changes, i.e., deviations in temperature and precipitation from average seasonal values due to anthropogenic activity [33]. All EU countries are recording extremes in their climate. The risk of aflatoxin contamination in cereals increases with increasing temperature for every 2 °C in EU countries, and a significant risk of increased incidence of maize contamination in the coming decades is estimated [34,35]. Namely, aflatoxin-producing species need temperature conditions of 25–37 °C and humidity of 80–85% for growth [36].

The Croatian Meteorological and Hydrological Service monitors the climatic characteristics of air temperature and precipitation in Croatia on the basis of average monthly, seasonal, and annual values and reports them as a representation of deviations from the multi-year average for the reference period 1981–2010. The summary annual reports showed that according to measured temperatures, the years 2017, 2018 and 2019 were: 2017, extremely warm in 85% of the territory of Croatia, and very warm in the remaining 15%; 2018, extremely warm in the entire territory; 2019, extremely warm in 50% of the territory and very warm for the remaining 50% [37,38,39].

Climatic characteristics recorded in 2019 were suitable for the development of mould and the synthesis of AFB_1_ in cereals, which was seen in the largest number of positive milk samples (38.5%) at the annual level. Spring 2019 (March–May), especially May, was characterised by a rainy to the very rainy state of precipitation, i.e., extremely rainy in central and southern Croatia in relation to the multi-year average (1981–2010) for Croatia. After May, extremely warm weather was recorded in June 2019 in central and eastern Croatia, with a temperature increase of 3.4 to 4.6 °C compared with average temperatures [40]. In addition, extremely warm weather was recorded in summer in central and southern Croatia (increase 2.1 to 2.9 °C compared with the average). The winter of 2019/2020 was also very warm. Given the above description of the climate during 2019, it is not surprising that 44.7% of all positive samples from central Croatia were recorded in 2019, with a higher incidence of positive samples measured during spring and summer (3.5 and 4.2%).

An increased number of positive samples were also found in the autumn and winter of 2021. The characteristics of the climatic conditions were increased temperatures in January and February in central and eastern Croatia with 1.8 to 3.9 °C higher temperatures than the average values and a heavier rainy season in January 2021. April and May 2021 were characterised by cold weather in central Croatia and very cold in eastern Croatia. May was very rainy in central Croatia. Extremely warm weather was recorded in June and July throughout Croatia and extremely dry in June. Furthermore, autumn September was warm and dry in central and eastern Croatia [40].

Extreme weather conditions, warm weather, and prolonged periods of drought, especially during maize growth and harvesting, were also recorded in previous years in Croatia, i.e., 2012. Such weather conditions facilitate fungal infection and stimulate the production of AFB_1_, resulting in contamination of cereals and dairy cow feed. A study conducted on corn samples during 2013 in Croatia (referred to the genus 2012) showed AFB_1_ concentrations higher than maximal permissible level in 28.8% of the samples. Therefore, it was concluded that higher AFB_1_ concentrations in grain mixtures and feed for cows could be attributed to a substantial AFB_1_ corn contamination determined in 2013 [18]. As a consequence of AFB_1_ contamination, high levels of AFM_1_ in milk were measured in Croatia [19]. This resulted in a crisis in 2013 when AFM_1_ levels exceeding the EU MRL were measured in 45.9% and 35.4% of milk samples collected in February and March 2013 in eastern Croatia [19] and further in 9.32% of samples in autumn 2013 [20]. The results obtained in this study were significantly lower than those found during 2013. However, with regard to territorial incidence of positive AFM_1_ levels in the present study, a higher frequency of elevated AFM_1_ concentrations in milk samples was recorded in central Croatia. A subsequent study in Croatian in 2014 showed a reduction in feed contamination with AFB_1_, i.e., lower concentrations of AFM_1_ (only 2.37% of milk samples exceeded EU MRL) were found in winter in eastern Croatia [20]. Further monitoring of AFM_1_ concentrations in Croatia showed a frequency of positive milk samples of 0.3% and 1.1% in spring and autumn 2016 [31].

The results of the frequency of AFM_1_ appearance in milk in the present study were also significantly lower than the results of different studies carried out in the period 2013–2018 in the neighbouring country of Serbia. A study conducted in the period 2013–2014 showed that 56.3% of raw milk samples exceeded EU MRL and the mean AFM_1_ was 358 ng/kg in winter and 375 ng/kg in spring [41]. Another study from Serbia conducted in the period 2013–2016 reported that 49.1% of milk samples had AFM_1_ levels exceeding the EU limits, with mean values between 153 and 353 ng/kg and a maximum of 5078 ng/kg measured in 2013 [42].

In Serbia during the period 2015–2018, 46.2% of raw milk samples exceeded the EU MRL, with the highest frequency of 65.4% and maximal mean level of 220 ng/kg in autumn [43]. Another study conducted in Serbia in 2015–2016 also showed that 30% samples exceeded the EU MRL, and in relation to the frequency of contamination during the seasons, those two years showed significant differences, i.e., the highest frequency in 2015 was determined in autumn (47%) and summer (22%) while in 2016 the highest was in winter (52.7%) and spring (33%) [44].

A significantly better situation was recorded in the neighbouring Italy during the period 2013–2018 when 31,702 milk samples were analysed and mean AFM_1_ concentrations were between 10.3 and 12.4 ng/kg, and concentrations exceeding the EU MRL were detected in only 0.2% of samples [45]. In addition, a study from Italy that monitored AFM_1_ concentrations in the period 2014–2020 showed a frequency of positive milk samples of 0.86%, while in 2015 the occurrence was 1.5% [46].

In a recent study from Albania conducted for the period 2019–2020, AFM_1_ exceeded the EU MRL in a total of 5.88% milk samples, and it was concluded that milk collected during 2019 had higher AFM_1_ levels with a maximal concentration of 217 ng/kg [47]. These results are similar to those obtained in this study, i.e., 2019 stands out as a year with a higher frequency of increased concentrations of AFM_1_ in milk compared with 2020.

Given the climatic predispositions for the development of toxigenic moulds and the production of AFB_1_ in tropical and subtropical countries of the world, the occurrence of AFM_1_ in milk is a constant phenomenon in these countries. Successive studies from Pakistan showed a significant incidence of raw milk contamination with AFM_1_ in both summer (36%, 19%) and winter season (40%, 29%) [14,48]. Studies from countries whose climatic and geographical features contribute to the increased incidence of AFM_1_ (> 50 ng/kg) in milk are Pakistan 53% [15], India 44% [49], Bangladesh 70% [50] and 23.8% [51], South Africa 81% [52], Ecuador 59.3% [53], and Ethiopia 62.5% [54]. The incidence of AFM_1_ in milk in countries whose legislation prescribes an MRL of 500 ng/kg showed an incidence above 500 ng/kg were Brazil 38% [55], Pakistan 69% and 90.9% [56,57], and Ethiopia 21.9% [54].

In this study, the highest AFM_1_ of 1100 ng/kg was measured in Central Croatia during September 2021 which suggests the usage of a highly contaminated feedstuff in the diet of dairy cows on that particular individual farm. Compared with the established value in a recent study from Pakistan, an extremely high mean AFM_1_ level of 1535.0 ng/kg was determined with a maximal value of 7460.7 ng/kg [57]. In addition, in a study from Brazil maximal concentrations of 3670 ng/kg were found in milk samples collected from December 2016 to November 2017 [55].

There are no literature data on the monitoring of AFB_1_ concentrations in corn or feed for dairy cows in Croatia for the period in which this study was conducted. According to previous studies, it can be concluded that the highest contribution to AFM_1_ in milk is due to contaminated corn, which is usually represented in the amount of 20–30% in feed concentrate used in the winter months for diet of dairy cow [18,58]. This is supported by a recently published study conducted for cereal samples from the 2017 harvest, which showed that corn samples were the most contaminated (8.7%) compared with wheat, barley, rye, and oat samples [59].

### 4.2. Exposure Assessment

According to EFSA, the main contributors to the total average exposure to AFM_1_ in all age groups of consumers are the food categories “liquid milk” and “fermented milk products” [28]. Assessing exposure to AFM_1_ allows scientific analysis to assess the severity and likelihood of adverse effects on human health through milk consumption, thus ensuring a link between possible hazards in the food chain and associated risks to human health. There is no consensus on the tolerable daily intake (TDI) for aflatoxins at the European Union level and therefore TDI is not determined for AFB_1_ or AFM_1_ [28]. Therefore, the EDI values found in this study were compared with the proposed TDI of 0.2 ng/kg BW/day [30].

In this study, the EDIs of positive AFM_1_ milk samples and the calculated HI index are presented in Table 3. The highest EDI values for AFM_1_ by season were determined for the autumn periods 2020 and 2021 and winter 2021/2022 according to the highest mean values of elevated AFM_1_ concentrations. The highest EDI of 1.15 ng/kg BW/day (5.74 ng/kg BW/day for 95th percentile milk consumption) was found in autumn 2021. Such high exposures were calculated taking into account only values above 50 ng/kg, which is in itself the worst-case scenario.

EFSA obtained chronic dietary exposure to AFM_1_ (ng/kg BW per day) for mean values and the 95th percentile dietary exposure between 0.04 and 0.06 ng/kg BW/day and 0.13 and 0.16 ng/kg BW/day in the total adult population of the European Union [28]. Compared with the EFSA, when all quantified AFM_1_ concentrations above the LOD are taken into account in this study, it can be concluded that a higher exposure of the adult population was found (0.18 ng/kg BW/day) compared with the average European values.

Different studies regarding the daily exposure to AFM_1_ conducted in Serbia have shown different results for adults, respectively, 0.503–1.420 ng/kg BW/day [60], 0.16–0.243 ng/kg BW/day [44], for students 1.238–2.674 ng/kg BW/day [61], 0.062–0.074 ng/kg BW/day [43]. For the Greek population, EDIs of 0.350–0.499 ng/kg BW/day were determined [61]. In a Brazilian study, the estimated daily intake of 0.10 ng/kg BW/day was found [62]. In Turkey, an EDI of 0.054 ng/kg BW/day was found [63]. For the Italian adult population, EDI values varied in the range 0.02–0.08 ng/kg BW/day and also 0.04–0.13 ng/kg BW/day for the large-portion-size consumers [45].

The EDI obtained in India showed a range between 0.034 and 1.036 ng/kg BW/day depending on three agro-climatic zones [1]. Dietary exposure of AFM_1_ in studies conducted in Pakistan also varied by season, i.e., 0.22–5.45 ng/kg/day during different seasons [15], 0.47–0.72 ng/kg BW/day in summer and 0.6–1.0 ng/kg BW/day in winter [48]. A recent study in Pakistan showed that the estimation of a mean AFM_1_ exposure for raw milk and processed milk (UHT and pasteurised) for consumers was 11.9 and 4.5 ng/kg BW/day, respectively [57].

In this study, the calculated HI < 1 was determined for mean milk samples with AFM_1_ between LOD and 49.9 ng/kg and also for the total mean for all quantified AFM_1_ values above the LOD for all seasons, which actually means that the average exposure to AFM_1_ in the observed five-year period does not pose a risk. However, if the concentrations are viewed by season and expressed according to positive AFM_1_ values, the determined HIs are significantly high (H > 1) and represent a health concern. In most seasons, HI was between 0.88 and 1.72. HI values were found to increase with seasonal positive mean AFM_1_ levels and EDIs and the highest HIs were calculated in autumn 2020 and 2021, and winter 2021/2022: 3.10 and 7.71, 5.74 and 14.1, and 3.1 and 7.62, respectively. The results showed that consumers who consume large amounts of milk (95th percentile milk consumption) were particularly exposed. It is known that chronic exposure to high concentrations of AFM_1_ increases the risk of liver cancer [28].

Similar results for HI were presented in Serbia for adults (over 26 years), in the range of 0.84–1.16 and values higher than 1 were found for females [44]. In a recent study from India, calculated HI values were above 1 in the range 1.05–5.18 [1]. In an Italian study conducted between 2013 and 2018, HI values were < 0.25 [45].

## 5. Conclusions

The presented results showed that the described weather conditions of extreme droughts contributed to the development of toxigenic moulds and influenced the increased frequency of milk contamination with AFM_1_ in the period when dairy cow receive substitute feed in autumn and winter. Given the influence of the seasons, it was found that during the autumn months AFM_1_ showed the highest mean values and statistically significant differences of means between years. Overall, the largest percentage of positive samples was found in central Croatia (69.7%). The obtained result is not surprising considering the climatic extremities, i.e., elevated temperatures and dry period to which region is exposed throughout all seasons. However, rates of milk contamination were not as high as in the crisis period during 2013–2014. Overall, according to the results obtained in this study, it can be concluded that the previous crisis has affected the overall improvement of the aflatoxins control system. This was particularly evident through increased official controls of AFB1 in feed or AFM1 in milk. In this way, there was a significant reduction of these natural contaminants in the food chain, thus ensuring the safety of milk and milk products in Croatia.

The risk assessment of AFM_1_ dietary exposure from raw cow milk indicates a high level of concern during autumn and winter from the public health perspective. The risk is particularly pronounced for consumers who consume large amounts of milk (95th percentile milk consumption). Given the regional exposure, it can be concluded that the population in central Croatia is more exposed to health risks due to AFM_1_ ingestion through milk consumption.

As climate change shows irreversible change and will increasingly affect feed production, there remains room for further improvements in feed processing and storage processes to reduce mycotoxin contamination during critical periods in autumn and winter. Future studies should focus on infants and young children, as this is the population considered most sensitive to AFM_1_ exposure due to the consumption of higher amounts of milk and dairy products.

## Figures and Tables

**Figure 1 foods-11-01959-f001:**
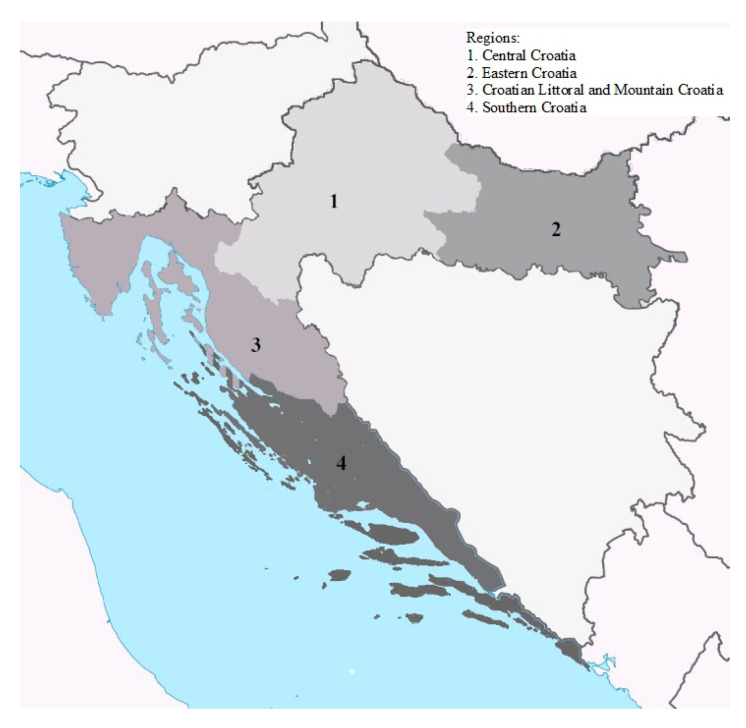
Map of Croatia with four indicated regions.

**Figure 2 foods-11-01959-f002:**
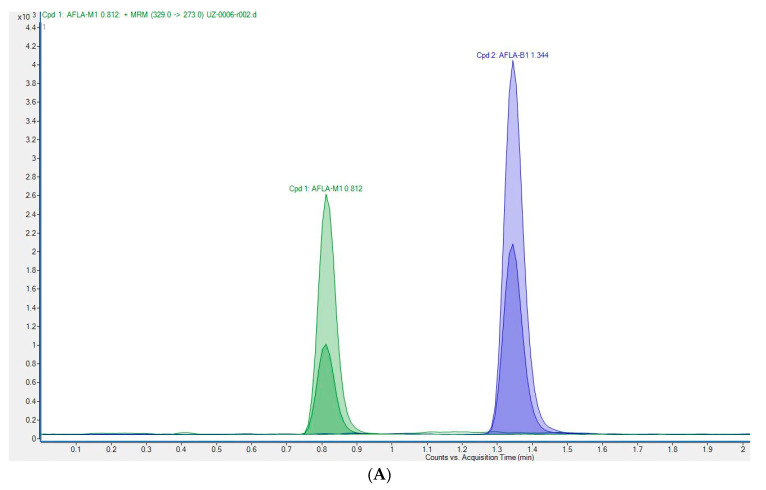
Chromatograms of AFM_1_: (**A**) spiked milk sample at 50 ng/kg; (**B**) positive milk sample (434.1 ng/kg).

**Table 1 foods-11-01959-t001:** MS/MS conditions for MRM analysis.

Mycotoxins	Precursor Ion (*m*/*z*)	Product Ion (*m*/*z*) *	Fragmentor (V)	Collision Energy (eV)	CAV	RT (min)
Aflatoxin M1	[M + H]^+^ 329	** 273.0 **	110	26	1	0.83
		259.2		24		
Aflatoxin B1 (internal standard)	[M + H]^+^ 313	** 284.9 **	110	26	1	1.38
		269.2		34		

* Quantifier ion (bold and underlined)—for quantification, qualifier ion—for compound confirmation.

**Table 2 foods-11-01959-t002:** Occurrence and distribution of AFM_1_ in raw cow samples collected in Croatia during five years period between winter 2016/2017 and winter 2021/2022.

Season	Total N	AFM_1_ Concentration (ng/kg) Distribution (ng/kg)			Positive Samples (≥50)	
		<LOD ^a^ *N* (%)	LOD—49.9 *N* (%)	≥50 *N* (%)	Range (ng/kg)	Mean ± SD (ng/kg)
Winter 2016/2017	482	472 (98.0)	4 (0.83)	6 (1.24)	52.2–85.4	67.1 ± 10.2
Spring 2017	207	205 (99.0)	1 (0.48)	1 (0.48)	62.1	
Summer 2017	135	133 (98.5)	0 (0)	2 (1.48)	72.7–79.3	76.0 ± 3.34
Autumn 2017	501	444 (88.6)	53 (10.6)	4 (0.80)	51.1–71.2	59.2 ± 7.44 *
Winter 2017/2018	660	612 (92.7)	42 (6.36)	6 (0.91)	52.5–79.3	66.7 ± 7.88
Spring 2018	308	304 (98.9)	3 (0.97)	1 (0.32)	87.0	
Summer 2018	205	195 (95.1)	10 (4.88)	0 (0)		
Autumn 2018	362	340 (93.9)	18 (4.97)	4 (1.11)	54.0–123.2	83.5 ± 25.9 *
Winter 2018/2019	305	288 (94.4)	10 (3.28)	7 (2.30)	54.6–87.1	74.1 ± 10.7
Spring 2019	229	219 (95.6)	2 (0.87)	8 (3.49)	54.1–81.6	71.3 ± 12.7
Summer 2019	263	243 (92.4)	9 (3.42)	11 (4.18)	50.7–114.5	88.9 ± 28.8
Autumn 2019	270	263 (97.4)	1 (0.37)	6 (2.22)	50.9–316.6	116.2 ± 79.9 *
Winter 2019/2020	250	219 (87.6)	15 (6.00)	16 (6.40)	50.3–122.7	77.9 ± 21.6
Spring 2020	146	141 (96.6)	4 (2.74)	1 (0.68)	135.1	
Summer 2020	237	230 (97.0)	4 (1.69)	3 (1.27)	80.2–88.2	85.8 ± 3.36
Autumn 2020	163	155 (95.1)	2 (1.22)	6 (3.68)	52.8–731.8	209.4 ± 237.6 *
Winter 2020/2021	346	333 (96.2)	2 (0.58)	11 (3.18)	51.0–92.0	70.4 ± 12.8
Spring 2021	124	123 (99.2)	1 (0.81)	0 (0)		
Summer 2021	162	158 (97.5)	1 (0.62)	3 (1.85)	50.7–72.6	61.3 ± 8.95
Autumn 2021	208	195 (93.8)	4 (1.92)	9 (4.32)	70.4 - 1100	387.8 ± 347.8 *
Winter 2021/2022	254	237 (93.3)	16 (6.30)	4 (1.57)	57.4–434.1	209.8 ± 152.9
Total winter	2297	2158 (93.9)	89 (3.87)	50 (2.18)	50.3–434.1	83.7 ± 59.7
Total spring	1014	992 (97.8)	11 (1.09)	11 (1.09)	54.1–135.1	77.7 ± 21.9
Total summer	1002	959 (95.7)	24 (2.40)	19 (1.90)	50.7–114.5	82.5 ± 24.4
Total autumn	1504	1397 (92.9)	78 (5.18)	29 (1.92)	50.9–1100	201.5 ± 253.3
Total	5817	5506 (94.7)	202 (3.47)	109 (1.87)	50.3–1100	116.1 ± 150.6

^a^ LOD 22.2 ng/kg [17]. * Significant differences in the AFM_1_ concentration between the different years.

**Table 3 foods-11-01959-t003:** Territorial incidence of positive raw cow milk samples collected in Croatia between winter 2016/2017 and winter 2021/2022.

Season	Total Positive N (>50 ng/kg)	Territorial Incidence of Positive Samples			
		Central Croatia	Eastern Croatia	Croatian Littoral and Mountainous Croatia	Southern Croatia
Winter 2016/2017	6	6	0	0	0
Spring 2017	1	1	0	0	0
Summer 2017	2	1	1	0	0
Autumn 2017	4	2	2	0	0
Winter 2017/2018	6	6	0	0	0
Spring 2018	1	1	0	0	0
Summer 2018	0	0	0	0	0
Autumn 2018	4	2	2	0	0
Winter 2018/2019	7	5	2	0	0
Spring 2019	8	6	1	0	1
Summer 2019	11	9	0	1	1
Autumn 2019	6	3	0	1	2
Winter 2019/2020	16	15	0	0	1
Spring 2020	1	1	0	0	0
Summer 2020	3	3	0	0	0
Autumn 2020	6	2	1	0	3
Winter 2020/2021	11	4	1	0	6
Spring 2021	0	0	0	0	0
Summer 2021	3	1	0	0	2
Autumn 2021	9	7	2	0	0
Winter 2020/2021	4	1	3		
Total Winter	50	37	6	0	7
Total Spring	11	9	1	0	1
Total Summer	19	14	1	1	3
Total Autumn	29	16	7	1	5
Total	109	76	15	2	16

**Table 4 foods-11-01959-t004:** Estimated daily intake (EDI) and Hazard Index (HI) for AFM_1_-positive samples for Croatian adults via consumption of raw milk during seasons 2016–2022.

Seasone	Estimated Daily Intake ^a^		Hazard Index ^d^	
	MC 1 ^b^	MC 2 ^c^	MC 1	MC 2
Winter 2016/2017	0.20	0.49	0.99	2.44
Spring 2017	-	-	-	-
Summer 2017	0.22	0.55	1.12	2.76
Autumn 2017	0.17	0.43	0.88	2.15
Winter 2017/2018	0.20	0.48	0.99	2.42
Spring 2018	-	-	-	-
Summer 2018	-	-	-	-
Autumn 2018	0.25	0.60	1.24	3.04
Winter 2018/2019	0.22	0.54	1.10	2.69
Spring 2019	0.21	0.52	1.06	2.59
Summer 2019	0.26	0.65	1.32	3.23
Autumn 2019	0.34	0.84	1.72	4.22
Winter 2019/2020	0.23	0.57	1.15	2.83
Spring 2020	-	-	-	-
Summer 2020	0.25	0.62	1.27	3.12
Autumn 2020	0.62	1.52	3.10	7.61
Winter 2020/2021	0.21	0.51	1.04	2.56
Spring 2021	-	-	-	-
Summer 2021	0.18	0.44	0.91	2.19
Autumn 2021	1.15	2.82	5.74	14.1
Winter 2021/2022	0.62	1.53	3.1	7.62
All seasons: mean for samples LOD–49.9 ng/kg ^e^	0.10	0.24	0.48	1.20
All seasons: mean for samples > LOD ^f^	0.18	0.45	0.91	2.23

^a^ Estimated daily intake, EDI (ng/kg bw/day). ^b^ MC 1 (mean milk consumption) = 2.96 g/kg bw/day [27]. ^c^ MC 2 (95th percentile milk consumption) = 7.27 g/kg bw/day (consumers consuming large milk quantities) [27]. ^d^ Hazard index, HI = EDI/TDI; TDI = 0.2 ng/kg bw/day [30]. ^e^ Mean = 33.1 ± 7.47 ng/kg. ^f^ Mean = 61.4 ± 99.5 ng/kg.

## Data Availability

The datasets generated for this study are available on request to the corresponding author.

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
