# Peer review of "Seasonal Occurrence of Aflatoxin M1 in Raw Milk during a Five-Year Period in Croatia: Dietary Exposure and Risk Assessment"

_foods, 2022, doi:10.3390/foods11131959_

Round 1
Reviewer 1 Report
This is valuable report on AFM1 occurrence in raw milk, as well as dietary exposure and risk assessment in Croatia with impressive number of analyzed samples. The paper is well-written, well-structured, clear, and concise. The methodology employed for mycotoxin determination, dietary exposure and risk assessment are novel and appropriate.
Title is suitable and illustrates the aim of the research.
Introduction is concise and informative, indicating the background and the aim if the research.
Methodology is described in detail, enabling the reader to replicate the research.
Comment 1:
Line 83-84. The authors indicated the sampling on farms with a smaller and larger number of dairy cows, but this was not discussed later. What is the significance of this information? Were the samples classified based on this in any way?
Comment 2:
The authors tested an impressive number of samples from two seasons. Was there a reason they did not deliver any statistical analysis of this seasonal effect?
Coincided, the effect of farm size could be interesting to be presented.
Results are presented in clear and understandable manner, and comprehensively discussed.
Conclusions are derived from the results and well-presented. However, since the authors have analyzed a great number of samples, more noteworthy conclusions could be obtained if deeper statistical analysis was carried out.
Author Response
Comment 1:
Line 83-84. The authors indicated the sampling on farms with a smaller and larger number of dairy cows, but this was not discussed later. What is the significance of this information? Were the samples classified based on this in any way?
Answer: a significant number of milk samples were collected from farms for which we did not have information on the size of the farm or the number of dairy cows. Namely, the laboratory used all the samples it received in the mentioned period for the AFM1 analysis. The laboratory received samples from authorized veterinary services and dairy industries and farms. Information on farm size was not available but only geographical position information was available. Since we did not have information on farm sizes, it is not planned to be included in the statistical processing.
Line 83-87 (2.1. Sample collection): according suggestions it is more clearly written regarding milk sampling: “In total, 5817 raw milk samples were collected in the period from winter 2016 to winter 2022. During that period, the laboratory received official samples of raw milk and raw milk from milk processing plants and farms sent within the self-control plans from all over Croatia. According to geographic information milk samples were sorted by sampling area in four geographical regions in Croatia (Figure 1) as well as according to the sampling season.”
Figure 1 is transferred to this chapter from the chapter 3. Results.
Comment 2:
The authors tested an impressive number of samples from two seasons. Was there a reason they did not deliver any statistical analysis of this seasonal effect?
Answer: a deeper statistical analysis was conducted in relation to the seasons. In Table 2 (Table 1 change to Table 2) were included total numbers and mean values by seasons. Statistical analysis was also performed with respect to the significant differences found for the positive concentrations determined in the different seasons
Correction for 2.5. Statistical Analysis: Statistical analysis was performed using: Small Stata 13.1 (StataCorp LP, 4905 Lakeway Drive, USA). The Shapiro-Wilk test was applied to determine the distribution of the data. Statistically significant differences (p < 0.05) between the same seasons between different years as well as between seasons in total were analysed by the Kruskal-Wallis test.
The determined results are inserted in the chapter 3. Results / 3.1. Occurrence of AFM1 in milk: “The order of the mean positive values with respect to the total values for the seasons is (lowest to highest): spring > summer > winter > autumn.
Statistical analysis showed significant differences (p=0.0313) in AFM1 concentrations for autumn seasons between the different years in which this study was conducted. However, no statistically significant differences (p > 0.05) were found for other three seasons between the different years. There were also no significant differences between the seasons considering their overall values throughout the study period.”
Statistical significant differences for autumn seasons were presented in Table 2.
Coincided, the effect of farm size could be interesting to be presented.
Answer: a significant number of samples came from farms for which we did not have information on the size of the farm or the number of dairy cows because, for example, only one sample came from the farm. The authors just stated that the samples were sampled on farms of different sizes with smaller and larger numbers of dairy cows. Since we did not have information on farm sizes, it is not planned to be included in the statistical processing. Therefore, in this study, the focus was on the sampling territory and the season.
Results are presented in clear and understandable manner, and comprehensively discussed.
Conclusions are derived from the results and well-presented. However, since the authors have analyzed a great number of samples, more noteworthy conclusions could be obtained if deeper statistical analysis was carried out.
Answer: deeper statistical analysis was carried out and incorporated to chapter 3. Results / 3.1. Occurrence of AFM1 in milk: “Statistical analysis showed significant differences (p=0.0313) in AFM1 concentrations for autumn seasons between the different years in which this study was conducted. However, no statistically significant differences (p > 0.05) were found for other three seasons between the different years. There were also no significant differences between the seasons considering their overall values throughout the study period.”
Explanation: regarding the noteworthy conclusions - the statistical results determined for the autumn only confirm the statements and conclusions that we have already stated in the discussion, ie “Seasonal variation can be explained by increased use of concentrated foods with higher amounts of mixed complementary foods such as dry hay and corn due to the reduced availability of fresh green feed in colder periods, thereby increasing exposure to feed contaminated with AFB1.” Etc…
- Correction for 2.5. Statistical Analysis: Statistical analysis was performed using: Small Stata 13.1 (StataCorp LP, 4905 Lakeway Drive, USA). The Shapiro-Wilk test was applied to determine the distribution of the data. Statistically significant differences (p < 0.05) between the same seasons between different years as well as between seasons in total were analysed by the Kruskal-Wallis test.
- The determined results are inserted in the chapter 3. Results / 3.1. Occurrence of AFM1 in milk: “The order of the mean positive values with respect to the total values for the seasons is (lowest to highest): spring > summer > winter > autumn. Statistical analysis showed significant differences (p=0.0313) in AFM1 concentrations for autumn seasons between the different years. However, no statistically significant differences (p > 0.05) were found for other three seasons between the years. There were also no significant differences between the seasons considering their overall values throughout the study period.”
- Statistical significant differences for autumn seasons were presented in Table 2.
- A conclusion has been added to the chapter 4. Discussion/ 4.1. Occurrence of AFM1 in milk: ”This was particularly emphasized in the autumn months in which contamination of milk with AFM1 showed the highest mean values and statistically significant differences of means between years.
- A conclusion has been added to the chapter 5. Conclusion: “Given the influence of the seasons, it was found that during the autumn months AFM1 showed the highest mean values and statistically significant differences of means between years.”

Reviewer 2 Report
The paper, titled “Seasonal Occurrence of Aflatoxin M1 in Raw Milk During a Five-Year Period in Croatia: Dietary Exposure and Risk Assessment”, looked at the seasonal occurrence of aflatoxin M1 (AFM1) in cow's milk from winter 2016 to winter 2022, as well as dietary exposure and risk assessment for the adult Croatian population. This is a methodical study involving a large quantity of data, and the findings of this research may be of interest to scientists, government officials, and the general public. This manuscript was also well-organized and well-debated. As a result, I believe this paper is appropriate for the journal Foods.
Here are some suggestions for improving the manuscript:
1. The number 1 in AFM1 is usually written as a subscript, AFM1.
2. Line 85: “The collected milk samples were grouped by seasons. ” seems not very accurate. The samples also include geographic information, according to the following text. As a result, I believe the milk samples were sorted by season and sampling area.
3. Line 154-157: the information on the mobile phase was incomplete. The elution procedure (phase A and phase B proportions) should be stated. However, the author need to check to see if the mobile phase run time is only 2 minutes. If so, chromatograms of AFM1 and IS of AFB1 should be provided.
4. Line 161-163: the information on MRM transitions should be provided.
5. Line 220: should “0.8-4.32” be expressed as “0.8-4.32%”?
6. Line 247-249: Table 1
① It is preferable to expand the statistical analysis for the complete seasons, as in Table 2.
② I propose that the authors conduct a separate analysis in the Discussion section for the extreme value (1100 ng/kg).
7. I think the authors should provide a literature review of AFB1 in cow's feed in the Discussion section. AFM1 in milk is known to come from AFB1 in feed. It's worth thinking about whether the findings of this article are consistent with the anticipation of AFB1 pollution in cow's feed.
Author Response
- The number 1 in AFM1 is usually written as a subscript, AFM1.
Answer: AFM1 is corrected to AFM1 in whole manuscript.
- Line 85: “The collected milk samples were grouped by seasons. ” seems not very accurate. The samples also include geographic information, according to the following text. As a result, I believe the milk samples were sorted by season and sampling area.
Answer: Line 83-87 (2.1. Sample collection): according suggestion it is more clearly written regarding milk sampling. Namely, the laboratory used all the samples it received in the mentioned period for the AFM1 analysis. Information on farm size was not available but only geographical position information was available.
Changes in the text are: “In total, 5817 raw milk samples were collected in the period from winter 2016 to winter 2022. During that period, the laboratory received official samples of raw milk and raw milk from milk processing plants and farms sent within the self-control plans from all over Croatia. According to geographic information milk samples were sorted by sampling area in four geographical regions in Croatia (Figure 1) as well as according to the sampling season.”
Figure 1 is transferred to this chapter from the chapter 3. Results.
- Line 154-157: the information on the mobile phase was incomplete. The elution procedure (phase A and phase B proportions) should be stated. However, the author need to check to see if the mobile phase run time is only 2 minutes. If so, chromatograms of AFM1 and IS of AFB1 should be provided.
Answer:
Line 160: the information on the mobile phase was added under chromatographic conditions: 60 % mobile phase A and 40 % mobile phase B.
Also, Table 1 is introduced in manuscript with MS/MS conditions for MRM analysis.
The chromatogram of spiked cow milk and positive milk sample is presented in text and as Figure 2 (in the penultimate paragraph of the chapter 2.3. Analytical Determinations).
- Line 161-163: the information on MRM transitions should be provided.
Answer: Table 1 is introduced in manuscript with MS/MS conditions for MRM analysis
- Line 220: should “0.8-4.32” be expressed as “0.8-4.32%”?
Answer: Line 220: 0.8-4.32 is corrected to 0.8-4.32% as also 0.91-6.4%
- Line 247-249: Table 1
① It is preferable to expand the statistical analysis for the complete seasons, as in Table 2.
Answer:
- A deeper statistical analysis was conducted in relation to the seasons. In Table 2 (Table 1 change to Table 2) were included total numbers and mean values by seasons. Statistical analysis was also performed with respect to the significant differences found for the positive concentrations determined in the different seasons
- Correction for 2.5. Statistical Analysis: Statistical analysis was performed using: Small Stata 13.1 (StataCorp LP, 4905 Lakeway Drive, USA). The Shapiro-Wilk test was applied to determine the distribution of the data. Statistically significant differences (p < 0.05) between the same seasons between different years as well as between seasons in total were analysed by the Kruskal-Wallis test.
- The determined results are inserted in the chapter 3. Results / 3.1. Occurrence of AFM1 in milk: “The order of the mean positive values with respect to the total values for the seasons is (lowest to highest): spring > summer > winter > autumn.
Statistical analysis showed significant differences (p=0.0313) in AFM1 concentrations for autumn seasons between the different years. However, no statistically significant differences (p > 0.05) were found for other three seasons between the years. There were also no significant differences between the seasons considering their overall values throughout the study period.”
- Statistical significant differences for autumn seasons were presented in Table 2.
- A conclusion has been added to the chapter 4. Discussion/ 4.1. Occurrence of AFM1 in milk: ”This was particularly emphasized in the autumn months in which contamination of milk with AFM1 showed the highest mean values and statistically significant differences of means between years.
- A conclusion has been added to the chapter 5. Conclusion: “Given the influence of the seasons, it was found that during the autumn months AFM1 showed the highest mean values and statistically significant differences of means between years.”
② I propose that the authors conduct a separate analysis in the Discussion section for the extreme value (1100 ng/kg).
Answer: in the Discussion added part related to that maximal value:
“In this study the highest AFM1 of 1100 ng/kg was measured in Central Croatia during September 2021 which suggests the usage of a highly contaminated feedstuff in the diet of dairy cows on that particular individual farm. Compared to the established value in recent study from Pakistan extremely high mean AFM1 level of 1535.0 ng/kg was determined with maximal value of 7460.7 ng/kg [57]. Also, in study from Brazil maximal concentrations of 3670 ng/kg was found in milk samples collected from December 2016 to November 2017 [55].”
- I think the authors should provide a literature review of AFB1 in cow's feed in the Discussion section. AFM1 in milk is known to come from AFB1 in feed. It's worth thinking about whether the findings of this article are consistent with the anticipation of AFB1 pollution in cow's feed.
Answer:
- To highlight what the reviewer states small modifications have been made to the Introduction Line 71-73: “Such climate change has affected the increased incidence of elevated AFB1 in dairy cattle feed [18] and appearance of elevated AFM1 concentrations in the milk and dairy products which was reported in countries such as Croatia [19,20], Serbia [21], Kosovo [22] or Macedonia [23].”
- As can be seen the explanations in relation to AFB1 contamination are already given in the end of the first paragraph of Chapter 4.1. Occurrence of AFM1 in milk: “Seasonal variation can be explained….
- Further addition to that is added to: „ Such weather conditions facilitate fungal infection and stimulate the production of AFB1 resulting in contamination of cereals and dairy cow feed. Study conducted on corn samples during 2013 in Croatia (referred to the genus 2012) showed AFB1 concentrations higher than maximal permissible level in 28.8% of the samples. Therefore it was concluded that higher AFB1 concentrations in grain mixtures and feed for cows can be attributed to a substantial AFB1 corn contamination determined in 2013 [18]. As a consequence of AFB1 contamination a high levels of AFM1 in milk were measured in Croatia [19].
- Also in the end of Chapter 4.1. Occurrence of AFM1 in milk is added paragraph:
“There are no literature data on the monitoring of AFB1 concentrations in corn or feed for dairy cow in Croatia for the period in which this study was conducted. According to previous studies, it can be concluded that the highest contribution to AFM1 in milk is due to contaminated corn, which is usually represented in the amount of 20-30% in feed concentrate used in the winter months for diet of dairy cow [18, 58]. This is supported by a recently published study conducted for cereal samples from the 2017 harvest, which showed that corn samples were the most contaminated (8.7%) compared to wheat, barley, rye and oats samples [59].”
- Two new references is added in reference list, and as a as a consequence, reference numbers had to be updated later in the text.
- The number 1 in AFM1 is usually written as a subscript, AFM1.
Answer: AFM1 is corrected to AFM1 in whole manuscript.
- Line 85: “The collected milk samples were grouped by seasons. ” seems not very accurate. The samples also include geographic information, according to the following text. As a result, I believe the milk samples were sorted by season and sampling area.
Answer: Line 83-87 (2.1. Sample collection): according suggestion it is more clearly written regarding milk sampling. Namely, the laboratory used all the samples it received in the mentioned period for the AFM1 analysis. Information on farm size was not available but only geographical position information was available.
Changes in the text are: “In total, 5817 raw milk samples were collected in the period from winter 2016 to winter 2022. During that period, the laboratory received official samples of raw milk and raw milk from milk processing plants and farms sent within the self-control plans from all over Croatia. According to geographic information milk samples were sorted by sampling area in four geographical regions in Croatia (Figure 1) as well as according to the sampling season.”
Figure 1 is transferred to this chapter from the chapter 3. Results.
- Line 154-157: the information on the mobile phase was incomplete. The elution procedure (phase A and phase B proportions) should be stated. However, the author need to check to see if the mobile phase run time is only 2 minutes. If so, chromatograms of AFM1 and IS of AFB1 should be provided.
Answer:
Line 160: the information on the mobile phase was added under chromatographic conditions: 60 % mobile phase A and 40 % mobile phase B.
Also, Table 1 is introduced in manuscript with MS/MS conditions for MRM analysis.
The chromatogram of spiked cow milk and positive milk sample is presented in text and as Figure 2 (in the penultimate paragraph of the chapter 2.3. Analytical Determinations).
- Line 161-163: the information on MRM transitions should be provided.
Answer: Table 1 is introduced in manuscript with MS/MS conditions for MRM analysis
- Line 220: should “0.8-4.32” be expressed as “0.8-4.32%”?
Answer: Line 220: 0.8-4.32 is corrected to 0.8-4.32% as also 0.91-6.4%
- Line 247-249: Table 1
① It is preferable to expand the statistical analysis for the complete seasons, as in Table 2.
Answer:
- A deeper statistical analysis was conducted in relation to the seasons. In Table 2 (Table 1 change to Table 2) were included total numbers and mean values by seasons. Statistical analysis was also performed with respect to the significant differences found for the positive concentrations determined in the different seasons
- Correction for 2.5. Statistical Analysis: Statistical analysis was performed using: Small Stata 13.1 (StataCorp LP, 4905 Lakeway Drive, USA). The Shapiro-Wilk test was applied to determine the distribution of the data. Statistically significant differences (p < 0.05) between the same seasons between different years as well as between seasons in total were analysed by the Kruskal-Wallis test.
- The determined results are inserted in the chapter 3. Results / 3.1. Occurrence of AFM1 in milk: “The order of the mean positive values with respect to the total values for the seasons is (lowest to highest): spring > summer > winter > autumn.
Statistical analysis showed significant differences (p=0.0313) in AFM1 concentrations for autumn seasons between the different years. However, no statistically significant differences (p > 0.05) were found for other three seasons between the years. There were also no significant differences between the seasons considering their overall values throughout the study period.”
- Statistical significant differences for autumn seasons were presented in Table 2.
- A conclusion has been added to the chapter 4. Discussion/ 4.1. Occurrence of AFM1 in milk: ”This was particularly emphasized in the autumn months in which contamination of milk with AFM1 showed the highest mean values and statistically significant differences of means between years.
- A conclusion has been added to the chapter 5. Conclusion: “Given the influence of the seasons, it was found that during the autumn months AFM1 showed the highest mean values and statistically significant differences of means between years.”
② I propose that the authors conduct a separate analysis in the Discussion section for the extreme value (1100 ng/kg).
Answer: in the Discussion added part related to that maximal value:
“In this study the highest AFM1 of 1100 ng/kg was measured in Central Croatia during September 2021 which suggests the usage of a highly contaminated feedstuff in the diet of dairy cows on that particular individual farm. Compared to the established value in recent study from Pakistan extremely high mean AFM1 level of 1535.0 ng/kg was determined with maximal value of 7460.7 ng/kg [57]. Also, in study from Brazil maximal concentrations of 3670 ng/kg was found in milk samples collected from December 2016 to November 2017 [55].”
- I think the authors should provide a literature review of AFB1 in cow's feed in the Discussion section. AFM1 in milk is known to come from AFB1 in feed. It's worth thinking about whether the findings of this article are consistent with the anticipation of AFB1 pollution in cow's feed.
Answer:
- To highlight what the reviewer states small modifications have been made to the Introduction Line 71-73: “Such climate change has affected the increased incidence of elevated AFB1 in dairy cattle feed [18] and appearance of elevated AFM1 concentrations in the milk and dairy products which was reported in countries such as Croatia [19,20], Serbia [21], Kosovo [22] or Macedonia [23].”
- As can be seen the explanations in relation to AFB1 contamination are already given in the end of the first paragraph of Chapter 4.1. Occurrence of AFM1 in milk: “Seasonal variation can be explained….
- Further addition to that is added to: „ Such weather conditions facilitate fungal infection and stimulate the production of AFB1 resulting in contamination of cereals and dairy cow feed. Study conducted on corn samples during 2013 in Croatia (referred to the genus 2012) showed AFB1 concentrations higher than maximal permissible level in 28.8% of the samples. Therefore it was concluded that higher AFB1 concentrations in grain mixtures and feed for cows can be attributed to a substantial AFB1 corn contamination determined in 2013 [18]. As a consequence of AFB1 contamination a high levels of AFM1 in milk were measured in Croatia [19].
- Also in the end of Chapter 4.1. Occurrence of AFM1 in milk is added paragraph:
“There are no literature data on the monitoring of AFB1 concentrations in corn or feed for dairy cow in Croatia for the period in which this study was conducted. According to previous studies, it can be concluded that the highest contribution to AFM1 in milk is due to contaminated corn, which is usually represented in the amount of 20-30% in feed concentrate used in the winter months for diet of dairy cow [18, 58]. This is supported by a recently published study conducted for cereal samples from the 2017 harvest, which showed that corn samples were the most contaminated (8.7%) compared to wheat, barley, rye and oats samples [59].”
- Two new references is added in reference list, and as a as a consequence, reference numbers had to be updated later in the text.

Reviewer 3 Report
The article is not innovative and similar studies have been conducted previously in Italy, Iran, Croatia, etc. A similar study was conducted in Croatia by Nina Bilandžić et al. In 2014-2015 entitled (Monitoring of aflatoxin M1 in raw milk over four seasons in Croatia), to reduce the production of this toxin, it is recommended that training and promotion programs for producers on the potential risk of aflatoxins reduce the concentration of AFB1 in livestock fodder. By observing strict hygiene regulations related to preparation and storage, as well as regular and continuous tests on raw milk samples, a practical study should be performed and the results should be analyzed.
Important points:
C1: In the sample collection section, the type of sampling must be specified. Is it a coincidence?
C2: The sampling method must be explained. For example, the health conditions during the sampling.
C3: In the introduction of the article, it was better to mention the advantages of using ELISA.
C4: Statistical analysis was better done with newer statistical software so that the reader has a better understanding of the article.
C5: In the abstract of the paper, for the purpose of confirming the concentration of AFM1 above the maximum permissible level of the European Union (MRL), high-performance liquid chromatography was performed by tandem mass spectrometry. It would be better to explain the accuracy of this method compared to the ELISA method in the introduction.
C6: This manuscript was a study estimating the seasonal occurrence of aflatoxin M1 (AFM1) in cow milk between winter 2016 and winter 2022 and assessing dietary exposure and risk assessment for the Croatian adult population. A total of 5817 bovine milk samples were screened for AFM1 concentration using an enzyme-linked immunosorbent assay (ELISA). And lacks any innovation.
C7: Keywords should be reviewed in relation to the title and purpose of the article
C8: For the purpose of confirming the concentration of AFM1 above the maximum permissible level of the European Union (MRL), high-performance liquid chromatography with mass spectrometry was performed on the samples. Chromatogram images need to be attached to the article.
C9: English The text of the article must be completely revised. Grammatical and lexical errors appear with a certain frequency. An example of them has been highlighted and corrected in the texts.
C10: Line 28 bw / day is stated for the first time, so it is necessary to include the full phrase.
C11: In this study, the concentration of aflatoxin M1 expressed in ng/kg is better expressed in ng/liter.

Author Response
Important points:
C1: In the sample collection section, the type of sampling must be specified. Is it a coincidence?
C2: The sampling method must be explained. For example, the health conditions during the sampling.
Answer: raw milk samples were collected from farms for which we did not have information on the size of the farm or the number of dairy cows. Namely, the laboratory used all the samples it received in the mentioned period for the AFM1 analysis.The laboratory received samples from authorized veterinary services and dairy industries and farms and therefore cannot state health conditions during the sampling.However, all samples were received in sterile bottles as described.
Line 83-87 (2.1. Sample collection): according suggestions it is more clearly written regarding milk sampling: “In total, 5817 raw milk samples were collected in the period from winter 2016 to winter 2022. During that period, the laboratory received official samples of raw milk and raw milk from milk processing plants and farms sent within the self-control plans from all over Croatia. According to geographic information milk samples were sorted by sampling area and season.
C3: In the introduction of the article, it was better to mention the advantages of using ELISA.
Answer: The ELISA method has been used and is used in a number of arlicles to determine AFM1 and there are a number of publications that have referred to its advantages and benefits. Since this is indeed very good and detailed described method, we believe that it is not necessary to state it so that the authors would not repeat it and state something so obvious and described many times.
C4: Statistical analysis was better done with newer statistical software so that the reader has a better understanding of the article.
Answer: Correction for 2.5. Statistical Analysis: Statistical analysis was performed using: Small Stata 13.1 (StataCorp LP, 4905 Lakeway Drive, USA). The Shapiro-Wilk test was applied to determine the distribution of the data. Statistically significant differences (p < 0.05) between the same seasons between different years as well as between seasons in total were analysed by the Kruskal-Wallis test.
C5: In the abstract of the paper, for the purpose of confirming the concentration of AFM1 above the maximum permissible level of the European Union (MRL), high-performance liquid chromatography was performed by tandem mass spectrometry. It would be better to explain the accuracy of this method compared to the ELISA method in the introduction.
C6: This manuscript was a study estimating the seasonal occurrence of aflatoxin M1 (AFM1) in cow milk between winter 2016 and winter 2022 and assessing dietary exposure and risk assessment for the Croatian adult population. A total of 5817 bovine milk samples were screened for AFM1 concentration using an enzyme-linked immunosorbent assay (ELISA). And lacks any innovation.
Answer: due respect to the reviewer's suggestions but the information requested to be included in the abstract is impossible to insert without transferring the default number of words of 200 which according to the default rules of Foods for the abstract. If this information were to be inserted, we would automatically have to remove information related to the results and conclusions, which in our opinion is more important than the suggested information. The abstract itself was shortened a lot from the original version to fit the 200-word quota.
C7: Keywords should be reviewed in relation to the title and purpose of the article
The authors think that the keywords are in line with the title
Answer: The authors think that the keywords are in line with the title:
Keywors are: aflatoxin M1; cow milk; public health; dietary exposure; seasonal exposure; risk assessment; Croatian regions
Title: Seasonal Occurrence of Aflatoxin M1 in Raw Milk During a Five-Year Period in Croatia: Dietary Exposure and Risk Assessment
C8: For the purpose of confirming the concentration of AFM1 above the maximum permissible level of the European Union (MRL), high-performance liquid chromatography with mass spectrometry was performed on the samples. Chromatogram images need to be attached to the article.
Answer: The chromatogram of spiked cow milk and positive milk sample is presented in text and as Figure 1 (in the penultimate paragraph of the chapter 2.3. Analytical Determinations).
C9: English The text of the article must be completely revised. Grammatical and lexical errors appear with a certain frequency. An example of them has been highlighted and corrected in the texts.
Explanation:English language is corrected by English speaker who is a native speaker (Grammatical and lexical errors were corrected and highlited in sent PDF).
In some cases no change has been made because:
Line 70: it says a moderate climate in European countries. Moderate climate is a term that cannot be given in the plural and describes the climate in the EU in general.)
Line 85: milk samples were collected throughout the year and then grouped by seasons
Lines 338-340: In sentence is stated by reviwer add an article but all sentense the whole sentence refers to this study and not to others („However, with regard to territorial incidence of positive AFM1 levels in the present study, a higher frequency of elevated AFM1 concentrations in milk samples was recorded in central Croatia.“)
C10: Line 28 bw / day is stated for the first time, so it is necessary to include the full phrase.
Answer: Line 28: bw/day is corrected to body weight/day
C11: In this study, the concentration of aflatoxin M1 expressed in ng/kg is better expressed in ng/liter.
Explanation: All results for study and in every day analysis in our laboratory were corrected from units ng/L to ng/kg due to the legal framework which determined that the MRL value was set as 50 ng/kg (density correction).

Round 2
Reviewer 3 Report
No further comments